# Integrating contextual sentiment analysis in collaborative recommender systems

**Nurul Aida Osman** [ID]**, Shahrul Azman Mohd Noah** [ID]**\*, Mohammad Darwich** [ID]**, Masnizah Mohd**

Center for Artificial Intelligence Technology, Faculty of Information Science & Technology, Universiti Kebangsaan Malaysia, National University of Malaysia, Bangi, Selangor, Malaysia

\* shahrul@ukm.edu.my

## Abstract

Recently. recommender systems have become a very crucial application in the online market and e-commerce as users are often astounded by choices and preferences and they need help finding what the best they are looking for. Recommender systems have proven to overcome information overload issues in the retrieval of information, but still suffer from persistent problems related to cold-start and data sparsity. On the flip side, sentiment analysis technique has been known in translating text and expressing user preferences. It is often used to help online businesses to observe customers' feedbacks on their products as well as try to understand customer needs and preferences. However, the current solution for embedding traditional sentiment analysis in recommender solutions seems to have limitations when involving multiple domains. Therefore, an issue called domain sensitivity should be addressed. In this paper, a sentiment-based model with contextual information for recommender system was proposed. A novel solution for domain sensitivity was proposed by applying a contextual information sentiment-based model for recommender systems. In evaluating the contributions of contextual information in sentiment-based recommendations, experiments were divided into standard rating model, standard sentiment model and contextual information model. Results showed that the proposed contextual information sentiment-based model illustrates better performance as compared to the traditional collaborative filtering approach.

## Introduction

Collaborative filtering technique is one of the most widely and applicable techniques used by recommender systems. The technique filters information by exploiting the rating of other similar users. The main concept of collaborative recommendation approaches is to exploit information about the past behaviour or the opinions of an existing user's community for predicting which items the current user of the system will most probably like [1]. Opinions that have currently been exploited by conventional recommender systems are usually in the form of numerical ratings. Collaborative filtering techniques perform well when there is sufficient rating information [2]. However, their effectiveness deteriorates when there exist insufficient

**Data Availability Statement:** All the data used during experiment are available from https://jmcauley.ucsd.edu/data/amazon/.

**Funding:** This study presented in the paper is supported by the Universiti Kebangsaan Malaysia's

Prime Impact Fund DIP-2020-017 awarded to Shahrul Azman Mohd Noah. There was no additional external funding received for this study.

**Competing interests:** The authors have declared that no competing interests exist.

ratings, which is a well-known problem in recommender systems called *data sparsity* [2]. Data sparsity leads to a general main problem, referred to as the cold start problem. Cold start is one of the long-standing problems in recommender systems [3] causes by the lack or no ratings on items by users. The aforementioned problems degrade the quality of recommender systems. Barragáns-Martínez et al. [4] addressed such problems using hybrid methods by combining collaborative and content-based systems. However, its applications are relatively limited to content-rich resources alone.

The vast growing of social media and e-commerce sites has led users to naturally post reviews or comments expressing their preferences or *feelings* towards items [5]. The comments are considered reliable indicators of users' overall satisfaction. The comments expressed by users may indicate a sentiment with different strength and intensity regarding a specific theme, expressing preferable, unpreferable or even neutral opinions. Thus, the opinions posted on recommender systems are useful for understanding the preferences that people have about items and products. Few research studies have considered sentiment analysis techniques in CF recommendations algorithms as a solution to address the issues data sparsity problem [6, 7].

Sentiment analysis is relatively a widely use technique for analysing user preferences and is potential to overcome some of the issues on CF recommendation algorithms. Employing such an analysis technique is seen capable of translating textual comments into explicit rating that can be processed by CF algorithm. Therefore, sentiment analysis technique is not only use to identify the sentences that contain different sentiment value [8] but also the strengths or intensity of such sentences. In the application of sentiment analysis techniques to CF, the approach taken by most of the research is by using the rating inference approach. This approach deduces numerical ratings from textual reviews, so that the user preferences expressed in the reviews can be integrated into CF algorithms. Such an approach is reflected in the works of [9–11].

Recommender systems currently suffer from a problem called domain sensitivity. The domain sensitivity problem in recommender systems can be surmounted by optimizing contextual information in conventional sentiment analysis models. In this sense employing the enhanced sentiment analysis technique with the contextual information able to reduce the ambiguity of opinion words in different domains, thus improving the overall recommendation quality.

Current sentiment-based recommender systems, however, do not consider contextual information. They are not able to accurately produce meaningful recommendations and precise information when dealing with multiple domains because of the existence of ambiguous words. The existence of ambiguous word involving multiple domains may influence the recommendation quality. For example, the ambiguous word '*horror*' will reflect a positive sentiment for a sentence in the movie domain such as *"This horror movie is really frightening. I can't wait for the next episode."*. While for a sentence in the product domain such as *"ABZ air cooler was totally not cool yet exclaimed in horror at the price."* reflects a negative sentiment. As a result, dealing with such contextual sentiment properties of opinion words is of importance for sentiment-based recommender systems.

This study, therefore, aims to examine the contribution of contextual information of sentiment analysis to recommender systems. The novelty of this work is by leveraging the technique of sentiment analysis and recommendation using collaborative filtering to generate a distinctive and functional contextual information-based recommender system. Therefore, this paper proposes a contextual sentiment-based recommendation framework to improve the recommender system services in the aspect of reducing data sparsity and error rate of the system performance.

## Background

### Recommender systems

Recommender Systems are software tools that give suggestions of items to be recommended to users. The suggestions aim to facilitate users in multiple decision-making processes, such as what items to purchase, what films to watch, or what books to read. recommender systems are used to produce good recommendations to a group of users for items or products that suit to their interest. Suggestions and recommendations for products on Amazon.com, and movies on IMDb, are real world examples of the online business operation of recommender systems. Recommender systems have proven to be a great solution to overcome the information overload and become one of the most powerful tools in online marketers. Recommender systems are usually grouped into three categories according to the approach of recommendations [1]:

1. Collaborative recommendation: CF is considered to be the best recommendation technique. Collaborative recommendation mimics the wisdom of the crowd idea where a user is recommended for items that are preferred by other 'similar' users in the past. CF recommendation works by utilizing user feedback in the form of ratings for items in a given domain and exploiting similarities in rating behaviour amongst several users in determining how to recommend an item. The main two approaches in CF are the item-based and the user-based [12]. Some early successes of CF on related domains included the GroupLen system [13]. CF is the most popular technique and algorithm in recommender systems. However, besides the reputation of CF, to a certain extent, it could not recognize the preferences of users in cold-start scenarios. Pan et al. [14] and Ghabayen and Noah [15] has proposed a solution to overcome the cold-start recommendation problem by exploiting social tags in their works. They used CF approach based on the expansion of users' tags. Tagging, for instance, has the potential of enhancing finding similar users, but to assume that users are interested with the items that were annotated is very naïve.

2. Content-based recommendation: While CF recommenders only utilize the user ratings matrix, content-based (CB) approaches treat all users and items as atomic single units. It works based on the data provided by users either explicitly or implicitly, which are then used to generate user profiles. In CB recommendation, items are recommended to a user based on the items the user liked in the past (stored as user's profile). CB filtering techniques largely rely on the information retrieval field, where the metadata and content of documents is used to select documents relevant to a user's query. In the context of recommender systems, recommendations are made by comparing representations of content describing an item to representations of content that interests a user [16]. CB recommendation can make a better personalized recommendation by knowing more about a user or an item, such as the actor, director, and genre of a movie [17].

3. Hybrid approaches: To elevate the strengths of CB and CF, hybrid approaches are being proposed. Producing separate ranked lists of recommendations of both approaches, and then merge their results to present a recommendation list [18] is the simplest form of hybrid. The other hybrid approaches are based on CF, but also maintain a CB profile for each user. Rather than co-rated items, the CB profiles are used to find similar users. Good et al. [19] used CF along with a number of personalized information filtering agents. Predictions for a user are determined by applying CF on the active user's and nearest neighbourhood user's personalized agents. Barragáns-Martínez et al. [4] also used a hybrid approach by combining CF and CB method enhanced with singular value decomposition to solve the issue of cold-start problem.

In CF recommender systems, ratings that described users' preferences on items, have been the most eminent representation to date [20]. Nevertheless, ratings have suffered from the long-standing problem called data sparsity when dealing with the large dimension of user-item matrix. As a result, few research works have taken place to resolve the above-mentioned problem [21]. One of the solutions is to utilize user preferences expressed in textual reviews, a problem known as sentiment analysis, and translate such preferences onto some rating scale that can be understood and processed by CF algorithms.

## Sentiment analysis

Sentiment analysis (SA) is widely applied in many domains, including commercial products and services, social media and customer relationship management [22]. SA generally analyses the structure of a textual review and infers it in the form of positive or negative sentiment.

Even so, to cope within the context of recommender systems, the value of sentiments must be translated into ratings scale. In other words, the statements expressed by users must be correlated in the form of numerical ratings. In sentiment analysis, one of the significant tasks is to recognize which words express a sentiment [6]. Words that are related with a sentiment intensity are also knows as opinion words. SentiWordNet dictionary is one of the popular linguistic resources in sentiment analysis in terms of providing answer to "how and which words people use to express preferences" [23]. The use of SentiWordNet has shown promising results in many studies as explained in [24]. Similar sentiment dictionaries have also been automatically created for other languages [25, 26].

Analysing the sentiments of movie reviews is one of the most studied area in sentiment analysis. Singh et al. [27] presented their experimental work on performance evaluation of the SentiWordNet approach for document-level sentiment classification of movie reviews and blog posts. Sahu and Ahuja [9] also implemented sentiment analysis task on IMDB movie review database by examining the sentiment expression to classify the polarity of movie reviews. Agarwal et al. [10] has done interesting work on sentiment analysis by using common-sense and contextual information. Their work has inspired us for the idea of combining sentiment analysis with collaborative recommendation by enhancing the sentiment analysis method into contextual based sentiment analysis method.

## Related works

The idea of exploiting opiniated textual reviews to overcome the long-standing problems of recommender systems mentioned earlier has been the main aim of this research. The sentiment analysis technique seems to be one of the worth-to-explore solutions by exploiting and making use of textual reviews. Research on sentiment analysis has shown that such reviews has the potential to generate better sentiment accuracy [10]. According to [11, 28], a sentiment analysis approach can be applied to textual reviews in order to infer users' preferences and subsequently map such preferences into some numerical ratings which can be understood and integrated into existing CF algorithm. Ricci et al. [29] also discussed the introduction and challenges in recommender systems, which has led to the solution of applying sentiment analysis in recommender systems. Some of the works that apply sentiment analysis in recommender systems are as follows.

The work by Guimarães et al. [30] use the polarity of adverbs to identify the polarity of sentences extracted from social networks. The solution applied in a recommendation system, which sends positive messages to users with sentences extracted from social network evaluated as negative polarity. Lei et al. [31] proposed to combine or fuse together three sentiment factors, mainly user sentiment similarity, interpersonal sentiment influence and item's reputation

similarity in order to improve the rating predictions of recommender systems. Another work worth mentioning is by Peleja et al. [8]. They proposed a recommender system for the TV on the web by integrating unrated reviews and movie ratings. In this case reviews of unrated movies are exploited by inferring ratings from the reviews and subsequently integrated with exiting user-ratings matrix. The work by Alhamid et al. [32] consider the physiological condition of user as a context parameter in order to improve recommendation results. Such a user's physiological context is based on the users' biological signals that determine their emotional situation. Thus, the ECG signal that measure the Heart Rate Variability (HRV) was used to detect the targeted physiological conditions of a user.

The aforementioned research explained the technique and solution applied to solve the general cold start and data sparsity problem in recommender systems. None of the works consider the ambiguity of words problem into their sentiment analysis methods. Words behave or meant differently based on the context or domain. Thus, it is a challenge to develop a universal opinion lexicon that has a polarity for every word. For example: The sentences "I love the vacuum cleaner that I purchased yesterday. It sucks very well" and "This movie is so bad. The actor is so suck!" show how context affects opinion word sentiment. In the first example, the word polarity of "sucks" is predicted as positive. In the second, the same word's polarity is negative.

Works that may look into ambiguity words problem is by Ziani et al. [33] that use the collaborative algorithm combined with features extraction and sentiment polarity classification to classify positive, negative and neutral sentiment. However, their word ambiguity problem is mainly concern on different languages presented in the reviews, which are Algeria: Arabic, Algerian dialect, French and English.

Wang et al. [34] proposed a movie recommendation framework based on a hybrid recommendation model and sentiment analysis on Spark platform. The work was proposed to improve the accuracy and timeliness of mobile movie recommender system. They implemented sentiment-enhanced recommendation framework by combining CF and CB methods. The review text was represented using VSM and TF-IDF, and the reviews were classified into positive and negative parts according to the sentiment lexicons.

Context is popularly referred as background, environment, setting, framework, or surroundings of events or occurrences. Within the scope of information seeking and retrieval, the time, location and past history of information access can shape the information needs of users. Thus, information system application needs to help users pull together data from disparate sources according to their expressed needs (as represented by system queries), as well as less specific criteria [35]. As such there are few works that proposed context-aware recommender systems (CARS). Haruna et al. [36] presented a review of recent developments in this area beginning January 2010 to October 2017 as a fountainhead for the research of a context-aware recommender system. The paper identified various application domains in the current recommender systems researches such as movies, music, and news which incorporate contextual information. Based on the review, contextual information is meant to identify places (spatial context), period of time (temporal context) or permanent (static context).

CARS mainly consider context as spatial, static and temporal. However, the contribution of this paper mainly focuses on the use of contextual words or phrases based on the domains which is one of the challenges in integrating reviews in recommender systems. The other three challenges are: sarcasm, types of negation, and multipolarity. The problem of word ambiguity is the difficulty to define polarity in advance because the polarity for some words is strongly reliant on the sentence context. For example, the word 'sucks' have different sentiment orientation for the domain of movie and electrical appliances. Therefore, the focus of this paper is more on the aspect of ambiguity of words. Such an aspect was not considered in the review

presented in [36]. The CARS has been researched extensively in various application domains such as in movies. Armentino et al. [37] built classification models for opinion mining to calculate sentiment based on the polarity of the target movies chosen in order to help users to decide which movies to watch. Their work did not integrate with any of the recommendation framework and no evaluative experiment has been conducted. Adomavicius & Tuzhilin [38] proposed to extend the recommendation capabilities by proposing the extension of 2-dimensional recommender system to multiple dimension such as time and space attribute for consideration.

The existing works of CARS approaches exhibit limitations in terms of semantic information. Therefore, Bollagella et al. [39] proposed a method to overcome this problem in a cross-domain sentiment classification by using a sentiment sensitive thesaurus. The cross-domain sentiment classification approach focuses on the ability of training a classifier from one or more domains (source domains) and applying a trained classifier to another domain that is the target domain. The created sentiment-sensitive thesaurus accurately captures words that express similar sentiments.

The aforementioned research, however, did not resolve the ambiguity problem where the authors mainly focused on improving the data sparsity issue, ignoring the precise content of the result presented to users. The lexicons used in the previous works are general lexicons, which are not appropriate for some domains. As a result, the recommender systems suffer in providing precise recommendation in different domains. The existence of ambiguous opinion words in different domains will lead to degraded recommendation quality. Thus, this paper proposed a novel enhanced-sentiment approach which is a domain sensitivity collaborative recommendation framework by integrating domain sensitivity information of sentiment analysis method in collaborative recommendation. Furthermore, within our knowledge, the extent to which the integration of inferred ratings from user reviews with the actual ratings enhanced the recommendation results is yet to be fully explored. Thus, in our experiment we consider different parameters of importance between the inferred ratings and actual ratings, in order to justify the contribution of both ratings in the overall recommendations.

## Methodology

The aim of this research is to improve the overall quality of collaborative recommender systems in terms of word ambiguity, data sparsity and error rate. One of the tasks is to integrate textual reviews into the user-item ratings matrix. By integrating textual reviews, data sparsity can be reduced, and subsequently improve the quality of recommendation. This study also aims to surmount the domain sensitivity problem in recommender systems by enhancing the sentiment analysis model with contextual information. Most of the existing approaches in sentiment-based recommender systems apply traditional recommendation techniques that lack of semantics information which lead to the major data sparsity and domain sensitivity problems. Hence, few have suggested the needs to consider domain sensitivity information in order to produce better sentiment accuracy and help to overcome data sparsity problem [10]. However, the extent of how potential domain sensitivity information can improve the quality of recommendation still remains a question.

Fig 1 conceptually illustrates the proposed solutions toward minimising the data sparseness problem by integrating the users' ratings on movies with textual reviews. As shown in user-item ratings, users $n_1$ and $n_2$ did not provide any rating to the movies of $i_2$ and $i_3$. However, instead of ratings, the users did leave some reviews about the movies. Here, sentiment analysis technique is performed to analyse the reviews. Textual reviews are decoded to some numerical rating on a scale of 1 to 5 by using a sentiment analysis technique. The generated sentiment

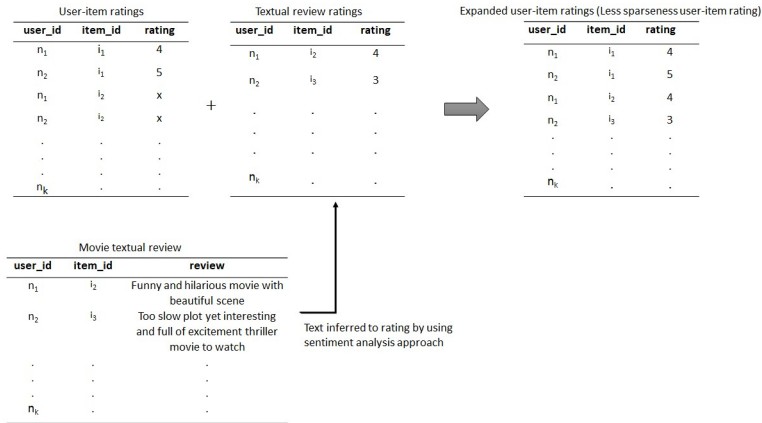

**Fig 1. Data sparseness solution by integrating the rating items and textual reviews.**

rating is then combined with the user-item ratings and produces the expanded user-item ratings, which are less sparse than user-item ratings. In this case, data sparseness can be improved by using the proposed solution.

The main reason why recommender systems failed to give good recommendations in different domains is due to the existence of ambiguous words. We refer to this as the domain sensitivity problem. For example, consider the ambiguous word "sucks". It will reflect a positive sentiment for a sentence in the product domain such as "*this vacuum sucks very well*", while for a sentence in the movie domain such as "*I hate this story. The actor's role is sucks.*" reflects a negative sentiment. Therefore, by adding contextual information based model to recommender framework, we expect such a problem to be resolved. Thus, embedding such enhanced-sentiment approaches are investigated and implemented in sentiment-based recommender systems. An overall flow process of contextual information based sentiment analysis in a recommender system is presented in Fig 2.

There are two main processes involved. First is the conversion of textual reviews into ratings. Second is the integration of sentiment rating to a collaborative filtering algorithm. Generally, the context-based sentiment analysis process infers textual reviews based on specific domain into numerical ratings format. Details of the process are shown in Fig 3. Furthermore,

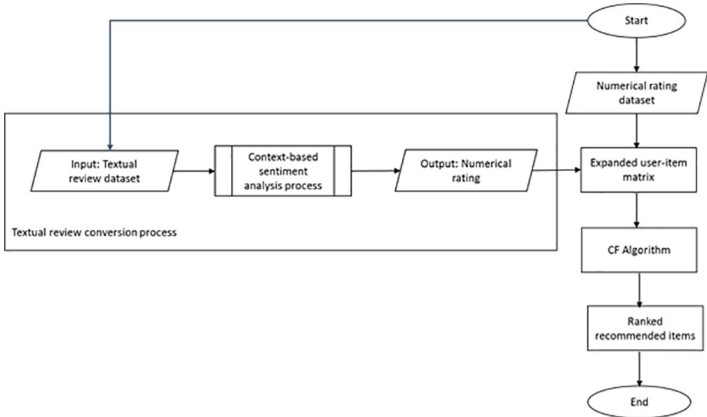

**Fig 2. An overall flow process of context-based sentiment analysis of collaborative recommendation.**

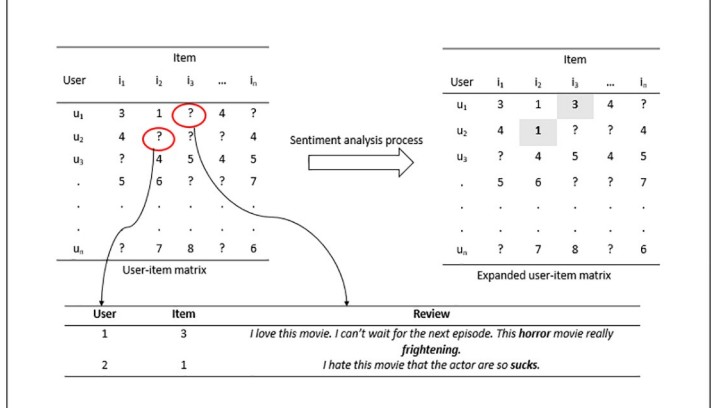

**Fig 3. The expanded user-item matrix filled with sentiment rating.**

the CF algorithm process calculates the recommendation value based on the expanded user-item matrix from the previous process. Both rating and sentiment rating are combined and embedded to CF algorithm process. In this experiment, for comparison purposes, the textual review dataset was taken from the Amazon dataset, from two different domains, which are TVs and Films series, and electronic products domain, consisting of 1000 reviews each. However, in this paper, we only include the experiment results based on the TVs and Films domain.

The process starts with analysing input from the dataset. There are two types of input dataset, which are the textual reviews and the numerical ratings datasets. For the input from the textual review, the conversion process first begins with the pre-processing texts. A rule-based approach is used to extract all the opinion words. The pre-processing task involves splitting text into sentences, part of speech (POS) tagging, and enhancing the POS-tagged text with our own tags terms. Reviews are then translated into numerical ratings by using conventional sentiment analysis technique. Sentiment analysis algorithm is used to analyse the reviews that users write about a movie and infer a preference in the form of ratings. A set of textual reviews and their associated ratings can be formulated as follows:

$$D = \{(re_{1,} nr_1) \ldots, (re_n, nr_{n,})\} \tag{1}$$

where a review $re_i$ is rated according to the value of rating $nr_i, \in \{1, 2, 3, 4, 5\}$. Reviews are represented by a set of opinion words (OW), as follows:

$$re = \{ow_{i,1}, \ldots, ow_{i,m}\} \tag{2}$$

where each element $ow_{i,m}$ represents the opinion word $m$ of the review $i$. An opinion word (OW) is a word that expresses a feeling or preference such as "interesting" or "disappointed". Furthermore, for each OW, the value of positiveness or negativeness was quantified. Therefore, the sentiment analysis algorithm aims at learning a classification function to infer the rating of the review $re_i$ as follows:

$$\Phi(re_i) \rightarrow [0, 1] \tag{3}$$

The function is learnt as a probabilistic model $p(ra_i, nr_i)$ estimated from a training set by following a machine learning technique. We used an unsupervised machine learning technique to compute the sentiment score for each review. The output from this process is a set of numerical rating data. Both input data are then merged to form an expanded user-item matrix.

Fig 3 shows the expanded user-item rating matrix filled with sentiment ratings, where the ratings reflect users' preferences. However, the ratings matrix is very sparse.

Using ratings alone is insufficient to make predictions and recommend reliable items to users. Therefore, contextual information in users' reviews are analysed by using sentiment analysis process. For example, user $u_1$ did not give any rating for item $i_3$, but did leave a comment, as shown in Fig 3. Sentiment analysis is used to analyse the comment and convert the textual comment into a numerical rating. Next, a numerical rating computed from sentiment analysis, called a sentiment rating, is filled into the expanded user-item matrix.

The recommendation process is typically computed using the user's feedback collected from all the respective users. It is normally represented in a user-item matrix. The entries in the user-item matrix are the ratings given by users on items. To determine the nearest neighbours in the neighbourhood-based algorithm, a subset of users is first chosen based on their highest similarity to the active user. The weighted combination of their ratings is then used to predict ratings for the active user. The similarity between two users is based on the cosine similarity measure.

Finally, items with a high prediction score is recommended and presented to user. The predictions are computed as the weighted average of deviations from the neighbours using collaborative recommendation calculation, as shown in Eq 4 as follows:

$$\hat{r}_{ui} = b_{ui} + \sum_{j \in D_c^k(u;i)} d_{ij}(r'_{uj} - b_{uj}) \tag{4}$$

where $\hat{r}_{ui}$ represents the rating of a user $u$ for an unknown item $i$. $b_{ui}$ is the estimation of the user $u$ for item $j$ and on a score calculated using the $k$ most similar items to $i$ that the user $u$ has already rated. The neighbourhood $D_c^k(u;i)$ is the neighbourhood of the $k$ most similar items that the user has commented. While $r'_{uj}$ is the result of the calculated function considering both explicit ratings and translated rating from comments.

Algorithm 1 explained the process of translating textual review into sentiment rating. The algorithm uses four word lists which are `NegationList`, `PositiveList`, `NegativeList` and `IntensifyList`. `PositiveList` and `NegativeList`, respectively, contain all the positive and negative lexicons. `NegationList` is a list of words use to negate the sentiment polarity of the sentence. The list contains the inverse lexicons such as *didn't*, *unlikely* and *can't*, whereas `IntensifyList` is a lost of words use to strengthen the sentiment polarity of the sentence. It includes increment terms such as *intensely* and *greatly* and decrement terms such as *hardly* and *barely*. These opinion lexicons are taken from Bing Liu lexicons [40].

Fig 4 depicts a flow diagram of enhanced-sentiment analysis approach in recommender system by implying the contextual information-based model.

As shown in Fig 4, the first process is to extract all potential opinion words (OW) from the corpus. All the opinion words were derived from the Stanford SNAP Movie Reviews corpus (https://snap.stanford.edu/data/web-Movies.html), which is an open access data.

In the second process, from the generated OW list, a set of predefined positive and negative seed terms are measured. The positive seed terms are *good*, *excellent*, *interesting*, and *happy*, whereas the negative seed terms are *bad*, *terrible*, *horrible*, and *poor*. A sentiment score for each OW in the range of [+1, -1] are generated using the measured list. The more accurate OW means the higher its sentiment strength.

In the third process, linguistic rules that consists of "OR" and "AND" rules are applied. The function of "AND" rule is used to join two adjectives of the OWs with similar sentiments while the "BUT" rule is used to join two adjectives with different sentiments. For all of the occurrences found between the seed words and OW AND POS_SEED_WORD, OW BUT

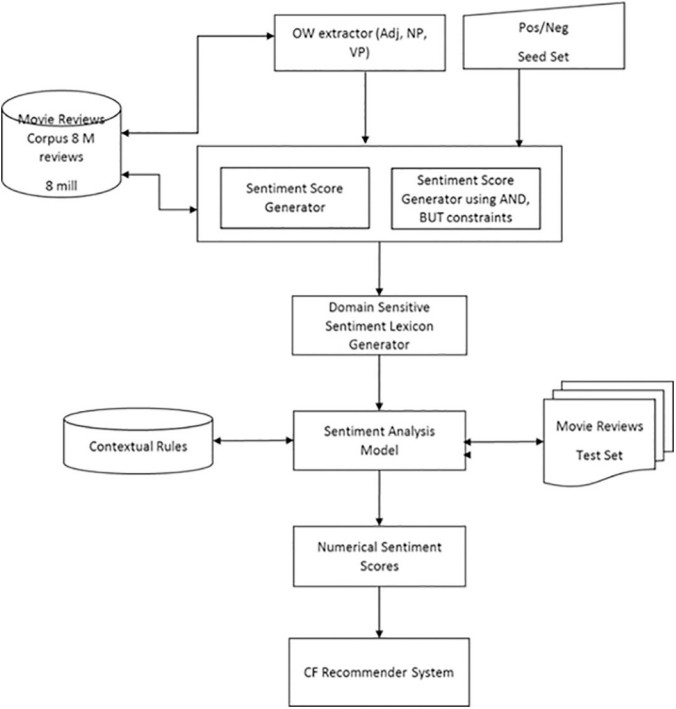

**Fig 4. The enhanced sentiment analysis process consisting of the following major components.**

POS_SEED_WORD, OW AND NEG_SEED_WORD, and OW BUT NEG_SEED_WORD from the corpus were taken. These occurrences within these linguistic constraints are calculated and a sentiment score from them are generated.

```
Algorithm 1. Pseudo-code algorithm for translating text into sentiment
rating
Input: Amazon product review
Output: Sentiment rating
Start
FOR every sentence in the product dataset:
   FOR each word in the sentence:
IF the word is in the NegationList
  NegativeNum = -1
ELSE:
  Expand word to SentimentList
  #Reset variables
  NegativeNum = 1
    FOR word in IntensifyList:
          IF word is in sentence:
             IF sentence is IncrementList
                PositiveCount = PositiveCount * 2
             IF sentence is DecrementList
                NegativeCount = NegativeCount / 2
     # Add as Positive sentence
     if positiveCount+negativeCount > accuracy:
                positive ++
                   Expand Array with sentiment score
     # Add as Negative sentence
```

```
        if positiveCount+negativeCount < -accuracy:
            negative ++
                Expand Array with sentiment score
End
```

The scores of adjectives from the second and third process are combined. Then the scores of noun and verb phrases generated in the second process are used. The generated sentiment lexicon is finely tuned and sensitive to the domain of the corpus. In order to improve the effectiveness of the proposed model, we performed the experiment on two different domains which are Amazon TVs and Films as well as electronic product. The sentiment lexicon generator combines the score from the two sentiment scores above it. It takes the average of each score for each word. The contextual rules are useful to improve overall accuracy. We also used valence shifter component such as negations, intensifiers and diminishers to generate more accurate results.

To evaluate the quality of this model, the accuracy value of the domain sensitivity generated lexicons was compared to the general lexicon. For the first experiment, we performed on the Movie Review dataset. Bing Liu's lexicon was chosen as a baseline model of the experiment. Bing Liu lexicon is a general domain independent lexicon used for sentiment classification. To demonstrate the performance between the models, the generated lexicon was then compared to general lexicon. We observed from the experiment that the generated lexicon performs better than the general lexicons because the proposed lexicon was compiled using a corpus of the same domain. For each review document, the lexicon to generate a final numerical sentiment score for it was used. The same experiment was performed on the electronic product review dataset. The accuracy value for both models are measured.

The implication of employing a domain-sensitive sentiment lexicon for the model was emphasized by the following real-world example. The sentence "I would suggest that you go read the book instead of watching the movie." would be labelled as having a neutral polarity by a general purpose lexicon, as it does not contain any explicit sentiment-carrying terms. However, using a domain sensitive lexicon that has been adapted to the movies domain, the sentence would be correctly flagged as a negative polarity, since the intention of the writer is to express a negative connotation towards the movie by pointing the reader to "*go read the book*" instead. Other terms labelled as negative in the movie domain are "*academic*" and "*predictable*". By observation, our domain-sensitive lexicon lists the term "book" as negative. This domain adaptation step would fine-tune the sentiment lexicon to tag terms with a polarity based on the domain itself, and in turn improve the overall accuracy of the sentiment analysis model.

After translating the textual into sentiment rating, the numerical scores inferred from the dataset documents can be used as the input for CF recommender system. CF algorithm would predict, rank and recommend movies to the user based on their sentiment toward each item. The Algorithm 2 described the process of generating scores of sentiment rating in textual reviews:

**Algorithm 2.** Pseudocode algorithm for generating sentiment scores within the linguistic constraints

```
Input: R0, PosRefTerm, NegRefTerm, review dataset
Output: R1 expanded
Start
PosConstraintScore_counter <-0
NegConstraintScore_counter <-0
For Synset in R0:
    For pos_term in PosRefTerms:
For each occurrence of "synset AND pos_term" OR "pos_term AND synset":
    PosConstraintScore_counter <- PosConstraintScore_counter + 1
      END For
```

```
For each occurences of "synset BUT pos_term" OR "pos_term BUT synset":
NCS_counter <- NCS_counter + 1
End For
For pos_term in NegRefTerms:
For each occurrence of "synset AND neg_term" OR "neg_term AND synset":
      NegConstraintScore_counter <- NegConstraintScore_counter + 1
END For
For each occurences of "synset BUT neg_term" OR "neg_term BUT synset":
PosConstraintScore_counter <- PosConstraintScore_counter + 1
End For
End For
End For
PosConstraintScore_counter <-0
NegConstraintScore_counter <-0
Polarity (synset)<-((PosConstraintScore-NegConstraintScore)/(PosCon-
straintScore + NegConstraintScore +1*10-5))
IF Polarity (synset) > 0:
        Polarity (synset) <- "positive"
ELSE IF Polarity (synset) < 0:
polarity (synset) <- "negative"
ELSE Polarity (synset) < "neutral"
END IF
S0 expand <- R1
End
```

In Algorithm 2, the final polarity score of the opinion word is calculated by dividing the value of subtracting positive and negative score of words with $10^{-5}$. To avoid the error "division by 0 error" in Algorithm 2, any small number were added to them. This is because positive constraint score (pcs) and negative constraint score (ncs) in the denominator are both 0. The small number was chosen so it does not affect the result of the final value.

Finally, the output of sentiment ratings from the previous process are integrated into the recommender system framework in order to establish a contextual information sentiment-based recommender model. The value of sentiment rating and numerical rating are then combined to get the overall rating as illustrated in in Eq 5:

$$P = \lambda ActRating + (1 - \lambda)SentRating \tag{5}$$

where *ActRating* is the actual rating provided by users and *SentRating* is the sentiment rating derived from users' reviews. $\lambda$ is a damping factor that decides the contribution of the individual ratings. During experiments we explore four values of $\delta$ which are 0.2, 0.4, 0.6 and 0.8.

## Experiments and results

The experiments were performed on standard, publicly available datasets, namely, the Amazon movie datasets containing rating and textual reviews (http://jmcauley.ucsd.edu/data/amazon/links.html) and the Amazon electronic products. We performed three different sets of experiments: the conventional ratings-based CF model; the enhanced CF model with sentiment ratings; and the CF model with contextual sentiment ratings. We called these models as ratings-based CF (ratingCF), sentiment CF (sentimentCF) and contextual sentiment CF (contextCF) respectively. The ratingCF and sentimentCF are a baseline for the experiment. In this experiment, we used Bing Liu lexicons as a baseline. The lexicons will later be expanded in the contextCF with the specific domain lexicon generated.

The datasets used in this experiment were derived from the Amazon benchmark dataset. To prove the effectiveness of the proposed method, the experiments were performed on two different domains. The baseline dataset consists of 50,000 ratings and 1000 textual reviews

(scaled between 1 and 5) from each TVs and Movies as well as electronic product domains. The datasets were then expanded by first adding generated sentiment ratings of movies using Algorithm 1, and second by adding generated contextual ratings of movies using Algorithm 2. A total of 1000 textual reviews from movies and electronic products domains were added into the baseline datasets. The experiment was performed on those three different sets of recommendation models. To guard against the data overfitting possibility, the data are divided into three parts which are the training data, evaluation data and testing data [41]. The datasets was divided into two portions, which are the training set that represents 80% of the original dataset, and the remaining 20% was used as the test set. The training set is used to train the recommendation model, while the test set has the items that used to be tested randomly for each user.

The performance of the suggestions provided by a recommender system should be measured by the value it can generate to the user as well as in terms of its predictive accuracy. One of the most fundamental measures for evaluating recommender systems is the accuracy value. The standard mean absolute error (MAE) and root-mean-square error (RMSE) are used to measure the performance of recommender system. RMSE and MAE mainly use to measure the accuracy of prediction and definitely the standard measure for evaluating recommender systems [1]. However, RMSE and MAE do not really replicate the real user experience. Thus, according to McLaughlin & Herlocker [42] precision and recall reflects the real user experience better than RMSE and MAE do, because in most cases, users actually received ranked lists from a recommender instead of predictions for ratings of specific items. As such precision and recall are measures that we used to evaluate value that a recommender system can generate to the user.

## Predictive accuracy

RMSE and MAE measure the value of error rate in recommendation. MAE is computed as follows [43]:

$$MAE = \frac{1}{N} \sum_{u,i} |P_{u,i} - r_{u,i}| \tag{6}$$

where $P_{u,i}$ is the predicted rating for user $u$ on item $i$, $r_{u,i}$ is the actual rating and $N$ is the total number of ratings on the item set. Lower readings of MAE indicate better prediction of user ratings by the recommendation engine. RMSE puts more importance on larger absolute error, and the lower the RMSE is, the better the recommendation accuracy. RMSE is given by Cotter et al. [18] as in Eq 7:

$$RMSE = \frac{1}{N} \sum_{u,i} (P_{u,i} - r_{u,i})^2 \tag{7}$$

Data sparsity level can be computed by using the Eq 8:

$$\text{Sparsity} = 100\% - \text{Density} \tag{8}$$

where;

$$\text{Density} = \frac{Number\ of\ Ratings\ in\ the\ Dataset}{(Number\ of\ movies) * (Number\ of\ Users)} * 100 \tag{9}$$

The sparsity value is opposed to the density value. The higher density level means the lesser the data sparsity should be. The overall value for sparsity can be obtained by subtracting the density value, which is detailed in Eq 9. The data sparsity value in the Amazon Tvs and Films

**Table 1. Results using different $\lambda$ weight.**

| $\lambda$ | sentimentCF | | contextCF | |
|---|---|---|---|---|
| | RMSE | MAE | RMSE | MAE |
| **0.2** | 3.68 | 4.54 | 3.60 | 4.40 |
| **0.4** | 3.64 | 4.50 | 3.58 | 4.38 |
| **0.6** | 3.60 | 4.48 | 3.53 | 4.36 |
| **0.8** | 3.57 | 4.43 | 3.45 | 4.33 |

dataset used is 90.80%. The quality of recommendations is highly reliant on the sparsity of available data [44]. The data sparsity levels in the electronic product dataset used is 99.80%. High sparsity levels pose a great challenge to the quality of predictions, and to the number of predictions computed [45]. In the common case, as the sparsity increases, CF algorithms may start to deteriorate as they are unable to form reliable neighbourhoods.

As the sentimentCF and contextCF are now involved with two ratings, which are the actual user ratings and generated sentiment ratings derived using algorithm 1 and algorithm 2, respectively, there is a need to assess which ratings contribute to the best performance. The results of initial experiments for four different values of $\lambda$ are illustrated in Table 1 (based on Eq 2). The results showed that the best performance was achieved for $\lambda = 0.8$, which indicates that the actRating contributes to the best result of both sentimentCF and contextCF. Hence, $\lambda = 0.8$ was used in the remaining experiments.

Table 2 shows the result of the experiment for all the three tests that has been conducted using Amazon TVs and Movies and the electronic product datasets review. The results show that the value of RMSE and MAE for sentimentCF model gives a minimal error compared to ratingCF for both domains. Further experiment was performed using the third test by optimizing CF with domain sensitivity sentiment analysis, called contextCF model to improve the RMSE, MAE and sparsity level. The results seemed to improve better and outperformed those previous tests by applying domain sensitivity sentiment analysis to the CF algorithm. The sparsity level also decreased from 90.80% to 90.70%, and 99.8% to 99.3% for both the TVs and Movies domain and the electronic products domain respectively. Overall, incorporating contextual information in sentiment-based collaborative recommendation outperform conventional the conventional collaborative recommendation approaches.

## User experience

As mentioned earlier, the performance metrics of precision and recall reflect the real user experience better than RMSE and MAE [42]. Both precision and recall are computed as fractions of $hits_u$ which is the number of correctly recommended relevant items for user $u$. The equations for precision ($P$) and recall ($R$) are as follows respectively, where $recset_u$ is the

**Table 2. Sparsity level, RMSE and MAE results for the three CF models.**

| Model | Amazon Tvs and Movies | | | Amazon Electronic Products | | |
|---|---|---|---|---|---|---|
| | Sparsity level (%) | RMSE | MAE | Sparsity level (%) | RMSE | MAE |
| **ratingCF** | 90.80 | 3.64 | 4.47 | 99.8 | 4.49 | 4.34 |
| **sentimentCF** | 90.80 | 3.57 | 4.43 | 99.7 | 4.04 | 3.80 |
| **contextCF** | 90.70 | 3.45 | 4.33 | 99.3 | 3.95 | 3.79 |

**Table 3. Precision at $k$ = 5, 10, 15 till K = 50 for three CF models.**

| Precision at k (P@k) | Amazon TVs and Films | | | Amazon Electronic products | | |
|---|---|---|---|---|---|---|
| | ratingCF | sentimentCF | contextCF | ratingCF | sentimentCF | contextCF |
| P@5 | 0.39 | 0.65 | 0.96 | 0.60 | 0.60 | 1.00 |
| P@10 | 0.53 | 0.61 | 0.87 | 0.70 | 0.70 | 1.00 |
| P@15 | 0.58 | 0.62 | 0.80 | 0.60 | 0.73 | 0.87 |
| P@20 | 0.63 | 0.65 | 0.70 | 0.55 | 0.75 | 0.85 |
| P@25 | 0.60 | 0.63 | 0.64 | 0.52 | 0.68 | 0.76 |
| P@30 | 0.66 | 0.67 | 0.70 | 0.50 | 0.67 | 0.73 |
| P@35 | 0.65 | 0.66 | 0.69 | 0.49 | 0.63 | 0.69 |
| P@40 | 0.63 | 0.67 | 0.70 | 0.48 | 0.60 | 0.68 |
| P@45 | 0.60 | 0.66 | 0.67 | 0.47 | 0.60 | 0.69 |
| P@50 | 0.56 | 0.61 | 0.64 | 0.41 | 0.56 | 0.68 |

recommended items for user $u$ and $testset_u$ refers to hits owing to the testing set size.

$$P_u = \frac{|hits_u|}{|recset_u|} \tag{10}$$

$$R_u = \frac{|hits_u|}{|testset_u|} \tag{11}$$

For evaluating models, precision and recall are useful metrics that produce binary results. Therefore, in recommender systems, we need to translate the numerical ratings output specifically from 1 to 5 into a binary format. In most research works, any true rating above 3.5 are assumed to be relevant and that any true rating below 3.5 is non-relevant. A relevant item means that this item is good for recommendation. Therefore, we will use this value as a general guideline in this works. The results for precision and recall are as shown in Tables 3 and 4 respectively. It is interesting to see that contextCF able to achieve better precision and recall values as compared to the other two CF models. The precision values suggest that the contextCF model able to rank and recommend relevant items to the users. The recall values also increased as the value of $k$ increased.

**Table 4. Recall at $k$ = 5, 10, 15 till K = 50 for three CF models.**

| Recall at k (R@k) | Amazon TVs and Films | | | Amazon Electronic products | | |
|---|---|---|---|---|---|---|
| | ratingCF | sentimentCF | contextCF | ratingCF | sentimentCF | contextCF |
| R@5 | 0.06 | 0.06 | 0.08 | 0.60 | 0.60 | 1.00 |
| R@10 | 0.14 | 0.12 | 0.16 | 0.70 | 0.70 | 1.00 |
| R@15 | 0.20 | 0.20 | 0.20 | 0.60 | 0.73 | 0.87 |
| R@20 | 0.26 | 0.26 | 0.28 | 0.55 | 0.75 | 0.85 |
| R@25 | 0.30 | 0.30 | 0.32 | 0.52 | 0.68 | 0.76 |
| R@30 | 0.40 | 0.40 | 0.42 | 0.50 | 0.67 | 0.73 |
| R@35 | 0.46 | 0.46 | 0.48 | 0.49 | 0.63 | 0.69 |
| R@40 | 0.50 | 0.54 | 0.56 | 0.48 | 0.60 | 0.68 |
| R@45 | 0.06 | 0.06 | 0.08 | 0.47 | 0.60 | 0.69 |
| R@50 | 0.14 | 0.12 | 0.16 | 0.41 | 0.56 | 0.68 |

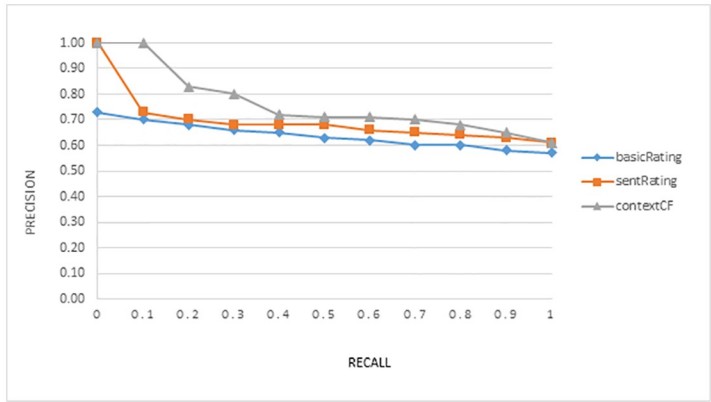

**Fig 5. Graph for average value of precision over recall for the experiment in the Amazon Tvs and Films domain.**

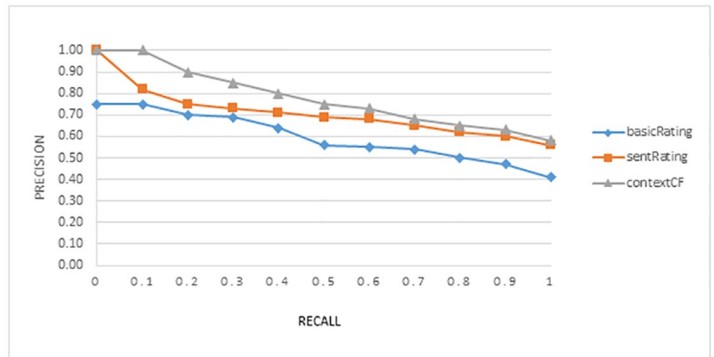

**Fig 6. Graph for average value of precision over recall for the experiment in the Amazon Electronic products domain.**

The average recall-precision graph in Figs 5 and 6 shows the trade-off between both measures and for both domains respectively. Both figures clearly show that incorporating contextual information achieved better performance as compared to the other two models.

The results of precision were further supported by the paired *t*-test results as illustrated in Tables 5 and 6. The tables show that the contextual-based recommender system approach (contextCF) significantly improved the recommendation quality in terms of precision.

## Threats to validity

Threats to validity of this research include external and internal threats. The recommendations produced any recommender systems are dependent on both the recommender algorithm as

**Table 5. Significance test for the Amazon Tvs and Films in terms precision.**

|  | ratingCF vs sentimentCF | | sentimentCF vs contextCF | | ratingCF vs contextCF | |
|---|---|---|---|---|---|---|
|  | **ratingCF** | **sentimentCF** | **sentimentCF** | **contextCF** | **ratingCF** | **contextCF** |
| Mean | 0.643 | 0.583 | 0.737 | 0.643 | 0.737 | 0.583 |
| t-Stat | 2.577 | | 2.641 | | 2.773 | |
| *P* | 0.030 | | 0.027 | | 0.022 | |

**Table 6. Significance test for the Amazon Electronic product in terms of precision.**

| | ratingCF vs sentimentCF | | sentimentCF vs contextCF | | ratingCF vs contextCF | |
|---|---|---|---|---|---|---|
| | **ratingCF** | **sentimentCF** | **sentimentCF** | **contextCF** | **ratingCF** | **contextCF** |
| Mean | 0.652 | 0.532 | 0.795 | 0.652 | 0.795 | 0.532 |
| t-Stat | 5.636 | | 3.956 | | 13.733 | |
| *P* | 0.0003 | | 0.003 | | 0.000 | |

well as the dataset used to train the system. In this experiment we only evaluate our proposed approach with a single dataset, thus the results may not generalize to other datasets. The dataset also may not be good representative of the whole population specifically since it comes from a single domain. Thus, it may be a threat to external validity of our experiments. Having said that, the proposed approach is general, and the movie dataset used on the experiment is commonly used and includes user and item ratings as well textual reviews relating to the items (i.e. movies) for collaborative-based recommendation.

In terms of distortion, the result is not distorted. However, the result may be distorted if a sentiment lexicon used for a specific domain was not properly built and validated. The lexicon generation for sentiment textual reviews must be well generated and fine-tuned according to the specific domain. For example, in the political domain, the textual review structure of sarcasm, slang, sentiment polarity and other attributes must be considered while applying sentiment analysis techniques.

As far as internal threats are concerned, our analysis was based on observation and evaluation accuracy by means of well-known measurement metrics: RMSE, MAE, precision and recall. These metrics do not allow much bias to be added to the analysis process [46], and thus pose minimal threat in this study.

## Discussion and conclusion

The results of the experiments clearly demonstrated that the proposed contextCF model outperforms the traditional sentimentCF and basic ratingCF models. To make it more domain specific and realistic, the experiments have been performed in different domains which is Amazon Tvs and Films and Amazon Electronic Products. The contextCF shows the lowest error rate value of RMSE and MAE compared to the other two in both domains. Our proposed solution also seems to slightly reduce the sparsity level from 90.80% to 90.70% in Amazon Tvs and Films and in Amazon Electronic Products it also reduces from 99.70 to 99.30. Furthermore, the proposed contextCF has shown to enhance the precision and recall values for $k = 5$, 10, 15 till $k = 50$. The proposed model has proved that the integration of contextual information sentiment-based model with collaborative recommendation can improve the recommendation quality. However, to fairly compare the proposed approach with other similar approaches seems unfeasible at the moment due to the language variations and differing data sets used during the experiment. The nearest similar approach is the one proposed by Ziani et al. [33], of which combination of opinion analysis and recommendation algorithm has shown to significantly improve the recommender system performance for Algerian users. The MAE values reported are 0.52, 0.50 and 0.60 for three sets of Algerian reviews and comments data sets respectively. Our proposed approach, however, exhibits better results.

The current sentiment-based recommendation approach still faced the issues of words ambiguity problem in different domains called domain sensitivity problem. This study is mainly to overcome the data sparsity and domain sensitivity issue in recommender systems. Current recommender systems fail to offer accurate recommendations in situation where

there is only few ratings are available. Applying contextual sentiment in recommender systems was seen to significantly improve the performance of recommender systems not only in terms of data sparsity as addressed by other sentiment-based recommender systems but also in terms of domain sensitivity caused by ambiguous words.

The problem of domain sensitivity or also knows as the ambiguous word in different domains was resolved with the aid of the contextual information sentiment based model. The contextual information sentiment-based model for recommender systems has managed to address the problem of domain sensitivity for situations where there exist ambiguous words in different domains. For example, the word "*unpredictable*" in the movie domain gives a positive sentiment for a sentence such as "*unpredictable plot*", while the same word gives a negative sentiment in the car domain, for a sentence such as "*unpredictable steering*". The proposed model which enhanced the traditional sentiment-based model has proven to gain better RMSE and MAE values, and has a reduced the percentage of data sparsity.

The results of the experiments also showed that actRating is still a better predicted element for both sentimentCF and contextCF. This is evident from the fact that the λ value of 0.7 gives better results as compared to the values of 0.3 and 0.5. It shows that actual rating still plays an important role in providing accurate recommendation. The findings are in line to opinion given by García-Cumbreras et al. [28]. Furthermore, the improvement of the contextCF approach is due to the expansion of the dataset with newly generated ratings for unrated items based on additional contextual information.

Other recommendation approaches such the content-based [47] and the graph-based approaches [48, 49] can benefit from the proposed method as these approaches may also exploit opinions of users in order overcome the long-standing data sparsity problem. The content-based approach may benefit the contextual information as the approach relies very much on textual information provided by users. While the graph-based approach that combines the content-based and collaborative-filtering approaches [48] may also be enhanced by integrating the contextual sentiment information in its algorithm.

## Future work

The proposed model has the potential to be applied to other domains as well. The lexicon can be generated with a fine-tuned lexicon to any other domains such as tourism [50], music [51] and citation recommendation [52], to produce an accurate and improvised recommender system. Future work includes the use of similar data sets of other previous works, so that a fair comparison can be made.

Apart from that, the proposed method also can be applied to other languages as well. The lexicon can be generated using Global WordNet Association that provides 50 type of languages in order to improve the recommendation content and quality. In this paper, the proposed method only deals with formal language. Therefore, it would be useful to enhance the language aspect that focuses on the slang and informal language to improve the recommendation quality.

## Author Contributions

**Conceptualization:** Shahrul Azman Mohd Noah.

**Data curation:** Masnizah Mohd.

**Formal analysis:** Nurul Aida Osman, Mohammad Darwich.

**Funding acquisition:** Shahrul Azman Mohd Noah.

**Methodology:** Nurul Aida Osman, Shahrul Azman Mohd Noah.

**Project administration:** Shahrul Azman Mohd Noah.

**Software:** Mohammad Darwich.

**Supervision:** Shahrul Azman Mohd Noah, Masnizah Mohd.

**Validation:** Mohammad Darwich, Masnizah Mohd.

**Writing – original draft:** Nurul Aida Osman.

**Writing – review & editing:** Nurul Aida Osman, Shahrul Azman Mohd Noah.

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
