## [Decision Letter · Decision Letter 0]

30 Nov 2020

PONE-D-20-33161

Integrating contextual sentiment analysis in collaborative recommender systems

PLOS ONE

Dear Dr. Noah,

Thank you for submitting your manuscript to PLOS ONE. After careful consideration, we feel that it has merit but does not fully meet PLOS ONE’s publication criteria as it currently stands. Therefore, we invite you to submit a revised version of the manuscript that addresses the points raised during the review process.

We look forward to receiving your revised manuscript.

Kind regards,

M. Usman Ashraf, Ph.D

Academic Editor

PLOS ONE

Journal Requirements:

"This research is partially supported by the Malaysia Ministry of Education Grant FRGS/1/2014/ICT02/UKM/01/1 awarded to Shahrul Azman Mohd Noah. The funder did not play any role in the study design, data collection and analysis, decision to publish, or preparation of the manuscript."

3. Thank you for submitting the above manuscript to PLOS ONE. During our internal evaluation of the manuscript, we found some occurrences of overlapping text with the following previous publication(s), some of which you are an author, which needs to be addressed:

- https://ieeexplore.ieee.org/document/8464694

- https://ieeexplore.ieee.org/document/7797377

- https://medium.com/@m_n_malaeb/recall-and-precision-at-k-for-recommender-systems-618483226c54

- https://www.toptal.com/deep-learning/4-sentiment-analysis-accuracy-traps

- https://www.rroij.com/open-access/real-time-sentiment-classification-usingunsupervised-reviews.php?aid=48846&view=mobile

- https://link.springer.com/book/10.1007%2F978-3-319-29659-3

We would like to make you aware that copying extracts from previous publications word-for-word, particularly outside the methods section, is unacceptable. In addition, the reproduction of text from published reports has implications for the copyright that may apply to the publications.

Please revise the manuscript to quote or rephrase the duplicated text and cite your sources for text outside the methods section. Please note that further consideration is dependent on the submission of a manuscript that addresses these concerns about the overlap in text with published work.

Reviewers' comments:

Reviewer's Responses to Questions

**Comments to the Author**

1. Is the manuscript technically sound, and do the data support the conclusions?

Reviewer #1: Yes

Reviewer #2: Partly

2. Has the statistical analysis been performed appropriately and rigorously? 

Reviewer #1: Yes

Reviewer #2: Yes

3. Have the authors made all data underlying the findings in their manuscript fully available?

Reviewer #1: Yes

Reviewer #2: Yes

4. Is the manuscript presented in an intelligible fashion and written in standard English?

Reviewer #1: Yes

Reviewer #2: Yes

5. Review Comments to the Author

Reviewer #1: Discuss Problem Statement clearly in manuscript, discuss more state-of-the-art work, Explain perfjoamcne metrics very clearly and justify, why you used the mentioned performance metrics in the manuscript, need to add more resulting graphs. add a new section named as Future Work, and discuss future work.

Reviewer #2: This paper describes a recommendation system approach

that leverages sentiment analysis to improve performance.

The following issues should be addressed before this paper

can be considered for publication:

-Table 1 could be replaced by a Figures with more values of distinct lambda.

- It is not clear if your results are significant better than other competing approaches.

- A better context on recommendation systems

should be provided. For example, it is not used

only for products. It can be used in wider contexts.

See and mention e.g. applications in collaboration and citation

analysis doi: 10.1209/0295-5075/99/48002

doi:10.1007/s00799-020-00288-2

- Figure quality should be much improved.

- there is no need to define cosine similarity.

- It would interesting to mention that other

strategies for recommendation system could also

benefit from the proposed strategy. For example,

graph based approaches could benefit from the

proposed sentiment analysis in recommendation systems.

See and mention e.g. doi: 10.1016/J.INS.2018.02.047

doi: 10.1145/544220.544231 for strategies that could be

combined and benefit from your approach.

- TAbles (in figure) should be provided in a standard (e.g. latex) way, and not

in a microsoft office figure.

- Perspectives for future works should be provided. Which other works

or lines of research could benefit from this strategy?

6. PLOS authors have the option to publish the peer review history of their article (what does this mean?). If published, this will include your full peer review and any attached files.

Reviewer #1: **Yes: **Dr. Mudassar Ahmad.

Reviewer #2: No

---

## [Author Response · Author response to Decision Letter 0]

18 Feb 2021

Editorial's comments

1. Please ensure that your manuscript meets PLOS ONE's style requirements, including those for file naming. The PLOS ONE style templates can be found at:

and

>>> Revised accordingly

"This research is partially supported by the Malaysia Ministry of Education Grant FRGS/1/2014/ICT02/UKM/01/1 awarded to Shahrul Azman Mohd Noah. The funder did not play any role in the study design, data collection and analysis, decision to publish, or preparation of the manuscript."

>>> The funding statement has been revised and updated

3. Thank you for submitting the above manuscript to PLOS ONE. During our internal evaluation of the manuscript, we found some occurrences of overlapping text with the following previous publication(s), some of which you are an author, which needs to be addressed:

- https://ieeexplore.ieee.org/document/8464694

- https://ieeexplore.ieee.org/document/7797377

- https://medium.com/@m_n_malaeb/recall-and-precision-at-k-for-recommender-systems-618483226c54

- https://www.toptal.com/deep-learning/4-sentiment-analysis-accuracy-traps

- https://www.rroij.com/open-access/real-time-sentiment-classification-usingunsupervised-reviews.php?aid=48846&view=mobile

- https://link.springer.com/book/10.1007%2F978-3-319-29659-3

We would like to make you aware that copying extracts from previous publications word-for-word, particularly outside the methods section, is unacceptable. In addition, the reproduction of text from published reports has implications for the copyright that may apply to the publications.

Please revise the manuscript to quote or rephrase the duplicated text and cite your sources for text outside the methods section. 

Please note that further consideration is dependent on the submission of a manuscript that addresses these concerns about the overlap in text with published work.

>>> Citation added and sentences been rephrased

4. Review Comments to the Author

Reviewer #1: 

a) Discuss Problem Statement clearly in manuscript, discuss more state-of-the-art work, 

b) Explain performance metrics very clearly and justify, why you used the mentioned performance metrics in the manuscript, need to add more resulting graphs. 

c) Add a new section named as Future Work, and discuss future work.

>>> Responses to Reviewer #1:

a. Problem statement and state-of-the-art work were further elaborated in the Introduction section.

b. Explanation of the used performance metrics were further elaborated and subsequently justified in the Experiments and Results section.

c. A new section named Future Work is included in the revised version of the paper

Reviewer #2: 

This paper describes a recommendation system approach

that leverages sentiment analysis to improve performance.

The following issues should be addressed before this paper can be considered for publication:

a) Table 1 could be replaced by a Figures with more values of distinct lambda.

b) It is not clear if your results are significant better than other competing approaches.

c) A better context on recommendation systems should be provided. For example, it is not used only for products. It can be used in wider contexts

See and mention e.g. applications in collaboration and citation analysis 

doi: 10.1209/0295-5075/99/48002

doi:10.1007/s00799-020-00288-2

d) Figure quality should be much improved.

e) There is no need to define cosine similarity.

f) It would interesting to mention that other

strategies for recommendation system could also

benefit from the proposed strategy. For example,

graph based approaches could benefit from the

proposed sentiment analysis in recommendation systems.

See and mention e.g.

doi: 10.1016/J.INS.2018.02.047

doi: 10.1145/544220.544231 

for strategies that could be combined and benefit from your approach.

g) TAbles (in figure) should be provided in a standard (e.g. latex) way, and not in a microsoft office figure.

h) Perspectives for future works should be provided. Which other works or lines of research could benefit from this strategy?

>>> Responses to Reviewer #2:

a) Noted. Table 1 has been replaces by a figures with more values of distinct lambda ranging from 0.2 till 0.8 to indicate more clear and specific results.

b) Results of significant test were included in Table 5 and 6. The results are significant better that the other two baselines which is traditional recommender system and traditional sentiment based recommender system. It has been proven in the experiments by comparing the results using the same format, same domain and same number of test sets.

c) In this experiment, we demonstrated in a wider context not only focus in one domain. We also use for movie and films domain not only for products. Hence, it can be used in different applications as well by performing in other domain such as medicine and political domains (See Future Works section).

d) The quality of figures have been improved.

e) The definition of Cosine similarity definition has been removed.

f) Other strategies that can be benefit from the proposed approach as recommended by the 2nd reviewer is included in the Discussion & Conclusion section.

g) Tables (in figures) has been changed in latex format.

h) Prospective for future work has been added in the Future Work section.

---

## [Editor Report · Decision Letter 1]

4 Mar 2021

Integrating contextual sentiment analysis in collaborative recommender systems

PONE-D-20-33161R1

Dear Dr. Noah,

We’re pleased to inform you that your manuscript has been judged scientifically suitable for publication and will be formally accepted for publication once it meets all outstanding technical requirements.

Kind regards,

M. Usman Ashraf, Ph.D

Academic Editor

PLOS ONE

---

## [Editor Report · Acceptance letter]

12 Mar 2021

PONE-D-20-33161R1 

Integrating contextual sentiment analysis in collaborative recommender systems 

Dear Dr. Noah:

I'm pleased to inform you that your manuscript has been deemed suitable for publication in PLOS ONE. Congratulations! Your manuscript is now with our production department. 

Kind regards, 

on behalf of

Dr. M. Usman Ashraf 

Academic Editor

PLOS ONE